# Comparison of Recovery Rates of Sudden Sensorineural Hearing Loss by Age Group

**DOI:** 10.3390/jcm13164937

**Published:** 2024-08-21

**Authors:** Yeo Ra Ha, Dokyoung Kim, Sung Soo Kim, Jae Min Lee, Sang Hoon Kim, Seung Geun Yeo

**Affiliations:** 1Department of Otorhinolaryngology Head & Neck Surgery, Kyung Hee University School of Medicine, Kyung Hee University Medical Center, Seoul 02447, Republic of Korea; yrha@khu.ac.kr (Y.R.H.); sunjaesa@hanmail.net (J.M.L.); shkim@khu.ac.kr (S.H.K.); 2Department of Anatomy and Neurobiology, College of Medicine, Kyung Hee University, Seoul 02447, Republic of Korea; dkim@khu.ac.kr; 3Department of Biochemistry and Molecular Biology, College of Medicine, Kyung Hee University, Seoul 02447, Republic of Korea; sgskim@khu.ac.kr

**Keywords:** sudden sensorineural hearing loss, age, recovery rate, comparison

## Abstract

**Background:** While numerous studies have been conducted on sudden sensorineural hearing loss (SSNHL), research on how treatment outcomes and prognosis vary by age remains insufficient. This study aims to investigate the differences in recovery rates among SSNHL patients divided by age groups. **Methods:** A retrospective study was conducted on 177 patients diagnosed with SSNHL from 2020 to 2023. The patients were categorized into four age groups: under 20, 30–40, 50–60, and over 70. Recovery rates and associated symptoms were compared across these groups. **Results:** Significant recovery rates were observed in all age groups except for those over 70 (*p* = 0.006, *p* = 0.003, *p* = 0.009). No significant differences were found in recovery rates based on gender (*p* > 0.75) or symptoms such as tinnitus, ear fullness, and dizziness (*p* > 0.05). **Conclusions:** The study revealed that younger and middle-aged adults showed statistically significant improvements in recovery rates, while the elderly exhibited relatively lower recovery rates.

## 1. Introduction

Sudden sensorineural hearing loss (SSNHL) is defined as a sudden onset of sensorineural hearing loss of 30 dB or more over at least three contiguous frequencies within three days. It is often accompanied by tinnitus, dizziness, ear fullness, nausea, and vomiting. SSNHL is one of the otologic emergencies that requires prompt treatment. It usually occurs suddenly in one ear, and while many cases recover, some do not fully recover, resulting in persistent hearing loss that can affect daily life [1]. Most cases do not have a discernible cause and have a variable prognosis, suggesting a multifactorial etiology. The primary suspected mechanisms include viral infections and vascular disorders, as well as cochlear membrane rupture, autoimmune diseases, head trauma, acoustic neuromas, and other causes [2,3].

Patients typically notice sudden hearing loss upon waking in the morning, although some may not recognize it until several days later. Some patients report fluctuating hearing loss, but most experience a gradual or rapid hearing loss after a sudden onset. In some cases, patients may first notice a feeling of ear fullness on the affected side before experiencing hearing loss. This ear fullness can occur with or without accompanying hearing loss. Additionally, about 90% of patients report varying degrees of tinnitus, which can precede the onset of SSNHL. Around 40% of patients experience dizziness or disequilibrium, with varying severity, but these symptoms usually resolve within a few days. Some patients also report otalgia or paresthesias [4,5,6].

SSNHL can occur at any age but is most commonly seen in individuals in their 40 s and 50 s, with a relatively lower incidence in the elderly and children [7]. The recovery rate of SSNHL may not always be favorable, prompting the proposal of various etiological analyses and treatment methods for a complete cure. This study aims to investigate the differences in the characteristics and prognosis of SSNHL across different age groups.

## 2. Materials and Methods

A retrospective review of medical records was conducted for 177 patients who visited the Department of Otolaryngology at K Hospital between 2020 and 2023 and were diagnosed with sudden sensorineural hearing loss (SSNHL) based on pure tone audiometry (PTA) and history. SSNHL was diagnosed in patients with previously normal hearing and no prior otologic disease who experienced a hearing loss of 30 dB or more over three contiguous frequencies within three days without an apparent cause. Patients were excluded if test results were insufficient or if they had bilateral hearing loss, fluctuating hearing, chronic inflammatory or suppurative ear disease or cholesteatoma, otosclerosis, prior ear surgery of any kind, auditory nerve tumors, cancer, trauma history, brain disease, Meniere’s disease, recent physical trauma or barotrauma to the ear, a strong family history of genetic hearing loss, craniofacial malformations, or acoustic neuroma. This study was approved by Kyung Hee University Hospital Institutional Review Board (IRB No 2023-04-060).

Hearing tests measure air and bone conduction thresholds, and the hearing threshold was calculated using the six-frequency average method for 500, 1000, 2000, and 4000 Hz (500 Hz + 1000 Hz × 2 + 2000 Hz × 2 + 4000 Hz/6). The severity of hearing loss was classified according to the 1964 ISO standard based on PTA: normal (<25 dB), mild (26–40 dB), moderate (41–55 dB), moderate–severe (56–70 dB), severe (71–90 dB), and profound (>91 dB). The audiogram types were classified as ascending (low frequencies have better thresholds than high frequencies), flat (similar thresholds across all frequencies), and descending (high frequencies have better thresholds than low frequencies). Audiogram configurations were classified as ascending (the average threshold of 250–500 Hz was 20 dB higher than at 4–8 kHz), descending (the average threshold of 4–8 kHz was 20 dB higher than at 250–500 Hz), or flat (similar threshold across the entire frequency range).

Hearing recovery rates were evaluated using Siegel’s criteria (Table 1): complete recovery (final hearing better than 25 dB), partial recovery (hearing improvement of 15 dB or more with final hearing thresholds between 25 and 45 dB), slight recovery (hearing improvement of 15 dB or more with final hearing worse than 45 dB), and no recovery (hearing improvement less than 15 dB or final hearing worse than 75 dB). Treatment involved high-dose oral steroids upon confirmed SSNHL diagnosis, with intratympanic steroid injections administered if there was no improvement. Steroid therapy consisted of 1 mg/kg/day of methylprednisolone for 4 days, followed by gradual tapering over the course of 2 weeks. Intratympanic steroid injections were administered at intervals for 3–4 days, twice a week, for a total of 3~8 times. The procedure was performed under a microscope, and 0.3–0.4 cc dexamethasone (Dexamethasone^®^, 5 mg/mL) was injected into the anterior inferior quadrant area of the tympanic membrane of the affected side using a 25-gauge spinal needle connected to a 1 cc syringe. To ensure sufficient contact of the injected dexamethasone with the round window, the supine position was maintained for 15–20 minutes with the affected ear facing up, and the patients were instructed not to swallow saliva or talk during this period.

Due to the limited sample size, age groups were categorized into ≤20 s, 20 s–40 s, 50 s–60 s, and ≥70 s groups. Treatment effects, associated symptoms, and recovery rates were compared among these age groups. Statistical analysis was performed using SPSS version 20.0 with independent *t*-tests, Wilcoxon rank-sum tests, ANOVA, and Kruskal–Wallis tests, with a significance level set at *p*-value < 0.05.

## 3. Results

### 3.1. Age and Gender Distribution

The age distribution was as follows: 9 patients aged 6–18 years, 13 patients in their 20 s, 14 patients in their 30 s, 26 patients in their 40 s, 53 patients in their 50 s, 39 patients in their 60 s, and 23 patients in their 70 s. For comparison, ages were categorized into four groups: ≤20 s, 30 s–40 s, 50 s–60 s, and ≥70 s. The average ages for these groups were ≤20 s (19.5 years), 30 s–40 s (41.9 years), 50 s–60 s (59.2 years), and ≥70 s (73.0 years). Among the 177 patients, there were 75 males and 102 females. There was no statistically significant difference in hearing recovery rates between genders (*p* = 0.759) (Table 2).

### 3.2. Recovery Rates by Age

Out of 177 patients, 153 (87%) experienced recovery: 59 (38.5%) had complete, 41 (26.7%) had partial, and 53 (34.6%) had slight recovery. Twenty-four patients (13%) showed no recovery. Significant recovery rates were observed in all age groups except for those over 70 (*p* = 0.006, *p* = 0.003, *p* = 0.009). The pre-treatment PTA averages were 58.3 dB (under 20), 57.9 dB (30–40), 64.2 dB (50–60), and 69.9 dB (over 70), with post-treatment averages of 35.6 dB, 38.1 dB, 45.9 dB, and 58.0 dB, respectively. The most significant recovery was noted in the 30–40 age group (*p* = 0.015) (Table 2).

### 3.3. Recovery Rates by Symptoms

Tinnitus was present in 120 out of the 177 patients, and there was no statistically significant difference in hearing recovery rates (*p* = 0.596). Ear fullness was present in 99 out of the 177 patients, and there was no statistically significant difference in hearing recovery rates (*p* = 0.826). Dizziness was present in 54 out of the 177 patients, and there was no statistically significant difference in hearing recovery rates (*p* = 0.419).

## 4. Discussion

In the United States, the prevalence of sudden sensorineural hearing loss (SSNHL) is reported to be 5–20 per 100,000 people, while in Korea, it is reported to be over 10 per 100,000 people. However, considering that the spontaneous recovery rate without treatment is reported to be about 32–65%, the actual prevalence is likely higher [8]. SSNHL can occur at any age, but it is most common in individuals in their 30 s to 50 s and shows no difference between genders. Additionally, approximately 90% of cases are unilateral, with bilateral involvement occurring in only 4–17% of cases, which is rare [9,10]. Particularly, the incidence in children is less than 10% of all SSNHL cases, and it is about 1.2% in those under 10 years old. In our study, it was 1.6% with three cases.

SSNHL usually recovers within two weeks, with approximately two-thirds of patients experiencing either complete or partial hearing recovery [11]. Additionally, recovery typically occurs early in the onset of symptoms, regardless of treatment, with about 78% of patients recovering within one month, 5.5% within two months, and a sharp decline in recovery rates after three months. However, once hearing starts to recover, it often improves rapidly within a few days [12]. In this study, patients diagnosed with SSNHL were immediately prescribed high-dose oral steroids. If there was no improvement with oral steroids, intratympanic steroids were administered 3–8 times. Consequently, the hearing test results in this study were based on hearing thresholds measured 2–3 months after the onset of hearing loss, showing similar outcomes to existing studies on treatment efficacy.

When SSNHL occurs, not all cases achieve full recovery, and some individuals experience persistent hearing loss that affects their daily lives permanently. Therefore, various mechanisms and treatments are being researched for a complete cure, but progress has been slow. Currently, known prognostic factors for SSNHL recovery include the initial degree of hearing loss, the configuration of the audiogram, the degree of speech discrimination, the presence of accompanying dizziness, tinnitus, and ear fullness, the degree of hearing loss at the start of treatment, the interval between onset and treatment initiation, the presence of underlying conditions, such as hypertension, diabetes, hyperlipidemia, metabolic disorders, other systemic diseases, medication compliance, and age. Additionally, many other factors can influence hearing recovery [1,2,13,14,15,16,17].

This study focused on comparing recovery rates by age group due to a lack of research on age among various factors. Most studies indicate that SSNHL occurring in older adults has a poor prognosis, whereas there is still debate about SSNHL in children. Recovery rates in children under 10 years old were extremely low at 25% [18]. In a study of 12 children with SSNHL under the age of 12, only one showed improvement, indicating poor prognosis in pediatric patients [19]. Similarly, poor prognosis was reported in children under 15 years old [20]. However, there have also been reports suggesting no relationship between age and treatment prognosis, and some studies have even shown higher recovery rates in children compared to adults [21,22]. Studies showing higher recovery rates suggest that, in the past, delayed detection and treatment due to communication difficulties in younger children made it challenging to diagnose SSNHL. However, with increased parental attention and early detection by parents and other family members, early treatment can lead to high recovery rates in children. The faster initiation of treatment is associated with higher recovery rates, as reported in numerous studies [23]. This underscores the necessity of prompt treatment for SSNHL in children regardless of age. In this study, there were only three children under the age of 10, precluding statistical analysis and comparison with other research findings. Among these three cases, one showed complete recovery, one showed partial recovery, and one showed no improvement.

Many studies have not reported better recovery rates in the elderly compared to younger groups; most studies have indicated either no relationship or lower recovery rates in older adults [1,22,24]. Several reasons contribute to poor prognosis in the elderly [25]. First, elderly patients may have presbycusis in both the affected ear and the healthy ear. Second, multiple comorbidities are often present in the elderly. Thus, conditions that can affect hearing may coexist with or precede the onset of SSNHL. Third, symptoms related to hearing loss in the elderly are atypical, and the accompanying symptoms are diverse. Fourth, chronic diseases, such as untreated hypertension, diabetes, hyperlipidemia, and metabolic disorders, are prevalent among the elderly. Fifth, the elderly have different responses to medications compared to younger individuals. They are more prone to side effects, which tend to last longer once they occur. Consequently, high-dose steroids used to treat SSNHL may be contraindicated, or their dosage and duration may need to be adjusted. Sixth, the elderly have compromised immune defenses, making treatment more difficult. The functions necessary to maintain homeostasis are diminished, and the occurrence of complications is higher. These factors likely contribute to the lower treatment efficacy observed in the elderly compared to children, young adults, and middle-aged individuals.

However, there are several limitations to this study:
Although the sample size is not small, it is somewhat insufficient for comparative observation across all age groups, including those under 10, teens, 20 s, 30 s, 40 s, 50 s, 60 s, and those over 70.As a retrospective study, it was difficult to accurately identify the factors affecting treatment and prognosis across different age groups without bias.Hearing typically begins to deteriorate in the late 30 s due to aging, and it was not possible to account for the variation in baseline hearing thresholds before the onset of SSNHL across different age groups.The Siegel criteria used in this study assess hearing outcomes based on the absolute hearing threshold of the affected ear without considering hearing in the opposite ear. This presents significant issues when applying these criteria to elderly patients, who are more likely to have some degree of hearing loss in the unaffected ear as well.


## 5. Conclusions

Although this study is retrospective and did not observe differences between age groups after excluding various pre-existing variables due to the large sample size, it did reveal that the recovery rate in young and middle-aged adults diagnosed with SSNHL is statistically significantly better. In contrast, the recovery rate in the elderly was lower.

## Figures and Tables

**Table 1 jcm-13-04937-t001:** Siegel’s criteria of hearing recovery.

Type	Hearing Recovery
I. COMPLETE RECOVERY	Final hearing better than 25 dB
II. PARTIAL RECOVERY	More than 15 dB gain, final hearing 25–45 dB
III. SLIGHT RECOVERY	More than 15 dB gain, final hearing 45 dB
IV. NO IMPROVEMENT	Less than 15 dB gain final hearing 75 dB

**Table 2 jcm-13-04937-t002:** Clinical characteristics of SSNHL patients.

	~20 (*n* = 22)	21~49 (*n* = 40)	50~69 (*n* = 92)	70~ (*n* = 23)	*p* Value
Age (Mean ± SD)	19.5 ± 6.3	41.9 ± 6.1	59.2 ± 5.4	73.0 ± 2.8	<0.0001
Sex (Male/Female)	8/14	22/18	37/54	8/16	0.281
Side (Right/Left)	10/12	21/19	41/50	9/15	0.703
Accompanying symptoms					
Tinnitus	14 (63.3%)	29 (72.5%)	61 (67.0)	16 (17.5%)	0.892
Ear fullness	15 (68.1%)	18 (45%)	56 (61.5%)	10 (10.9%)	0.618
Vertigo	5 (22.7%)	11 (27.5%)	30 (32.9%)	8 (8.7%)	0.048
Type of audiogram					
Ascending	4 (18.1%)	12 (30%)	12 (13.1%)	2 (8.3%)	0.937
Descending	4 (18.1%)	12 (30%)	21 (23.0%)	6 (25%)	0.618
Flat	10 (45.4%)	8 (20%)	34 (37.3%)	10 (41.6%)	0.316
Degree of deafness					
Slight	4 (18.1%)	3 (7.5%)	7 (7.6%)	0 (0%)	0.269
Mild	4 (18.1%)	11 (27.5%)	15 (16.4%)	3 (12.5%)	0.319
Moderate	3 (13.6%)	5 (12.5%)	18 (19.7%)	4 (16.6%)	0.296
Moderately Severe	3 (13.6%)	10 (25%)	12 (13.1%)	4 (16.6%)	0.436
Severe	3 (13.6%)	4 (10%)	20 (21.9%)	8 (33.3%)	0.379
Profound	5 (22.7%)	7 (17.5%)	19 (20.8%)	5 (20.8%)	0.492

Mean ± SD, dB.

## Data Availability

Data presented in this study are available upon request from the corresponding author.

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
