# Peer review of "Comparison of Recovery Rates of Sudden Sensorineural Hearing Loss by Age Group"

_jcm, 2024, doi:10.3390/jcm13164937_

Round 1

Reviewer 1 Report

Comments and Suggestions for Authors

Even taking into account the limitations of a retrospective study,  I consider that the aim of the study was not fully reached. "This study aims to investigate the differences in characteristics and prognosis of SSNHL across different age groups".

- no clear inclusion/ exclusion criteria

-no clear description of audiogram types: "classified as ascending (low frequencies have better thresholds than high frequencies)"

- in Table 2.  gender distribution within the study groups is present, but no data regarding hearing recovery. 

-how were the accompanying symptoms evaluated in order to apply statistical evaluation?

-there are no p values for descending and flat type audiogram, as well as for the degrees of deafness.

- there is no mention regarding the type of audiogram and the recovery prognosis.

Reviewer 2 Report

Comments and Suggestions for Authors

It would be interesting to know the time delay from symptoms onset to therapy by age groups. It would give more weight to results and present good ground for discussion.

More data are required about administered therapy (doses, oral or parentheral aplication, concentration of corticosteroids injected intratympanicaly, decision criteria for the number of shots. Children were obviously not treated with intratympanic aplications.

All coments are given in the provided file.
